# High-Performance Soy Protein Isolate-Based Film Synergistically Enhanced by Waterborne Epoxy and Mussel-Inspired Poly(dopamine)-Decorated Silk Fiber

**DOI:** 10.3390/polym11101536

**Published:** 2019-09-20

**Authors:** Huiwen Pang, Shujun Zhao, Tao Qin, Shifeng Zhang, Jianzhang Li

**Affiliations:** 1MOE Key Laboratory of Wooden Material Science and Application, Beijing Forestry University, No. 35 Tsinghua East Road, Haidian District, Beijing 100083, China; panghuiwen1010@bjfu.edu.cn (H.P.); blzhaoshujun@bjfu.edu.cn (S.Z.); blqintao@163.com (T.Q.); lijzh@bjfu.edu.cn (J.L.); 2Beijing Key Laboratory of Wood Science and Engineering, Beijing Forestry University, No. 35 Tsinghua East Road, Haidian District, Beijing 100083, China

**Keywords:** soy protein, waterborne epoxy, silk fiber, mussel-inspired, dopamine

## Abstract

It remains a great challenge to fabricate bio-based soy protein isolate (SPI) composite film with both favorable water resistance and excellent mechanical performance. In this study, waterborne epoxy emulsions (WEU), which are low-cost epoxy crosslinkers, together with mussel-inspired dopamine-decorated silk fiber (PSF), were used to synergistically improve the water resistance and mechanical properties of SPI-based film. A stable crosslinking network was generated in SPI-based films via multiple physical and chemical combinations of WEU, PSF, and soy protein matrixes, and was confirmed by attenuated total reflectance-Fourier transform infrared (ATR-FTIR) spectroscopy, X-ray diffraction (XRD), and solid state ^13^C nuclear magnetic resonance (^13^C NMR). As expected, remarkable improvement in both water resistance and Young’s modulus (up to 370%) was simultaneously achieved in SPI-based film. The fabricated SPI-based film also exhibited favorable thermostability. This study could provide a simple and environmentally friendly approach to fabricate high-performance SPI-based film composites in food packaging, food preservation, and additive carrier fields.

## 1. Introduction

Due to environment disturbances and the envisaged future shortfall of petroleum and petroleum-derived products, interest has been ignited for the development of environmentally friendly materials from renewable and biodegradable resources such as proteins, polysaccharides, and lipids [1,2,3,4]. In particular, soy protein isolate (SPI)-based film, a bio-based material with abundant sources and good biodegradability and biocompatibility, shows to be more promising than petroleum-based material [5,6]. However, poor water resistance and low mechanical strength are two major obstacles that impede the use of SPI-based film for extensive practical application [7,8,9]. Previous reports have developed various strategies to solve these two problems via chemical grafting [5,10], physical blending [8], crosslinking [11,12,13], and enzymatic treatment [14,15], and chemical crosslinking has been proven to be the most efficient approach for the improvement of SPI-based film [16].

Among various crosslinkers, epoxy group-containing crosslinkers have admirable effects on the mechanical properties and water resistance of SPI-based film [17]. This is due to the formation of a dense crosslinking network via multiple reactions between active epoxy groups and polar groups, such as –OH and –NH_2_, of the soy protein molecule [18]. Common epoxy crosslinkers are mainly epoxy diluents, such as ethylene glycol, di-glycidyl ether, and 1,2,3-propanetriol-diglycidyl-ether. Though the performance of SPI-based film is obviously improved with the introduction of epoxy diluents, their high cost and heterogeneous dispersion in the matrix impede their practical large-scale application.

Epoxy resins, as low-cost industrial adhesives, are widely used in the adhesive and coating industries due to their high reactivity and the excellent mechanical properties of the cured product [19,20]. However, epoxy resins are non-polar polymers [21], and have poor compatibility with the polar side chain of the soy protein molecule, which makes them unsuitable candidates for SPI-based film modification. Transforming non-polar polymers into waterborne and water-soluble matrixes was a popular approach in the fields of adhesive, coating, and composite production [22,23]. Waterborne matrices are characterized by a high level of compatibility with hydrophilic matrices, a relatively low cost, and good reactivity [24]. This provides a new pathway for transforming non-polar epoxy resin into a waterborne epoxy emulsions (WEU) matrix characterized by good compatibility with the polar side chain of the soy protein molecule, and makes it possible for low-cost waterborne epoxy resin to be used in the crosslinking modification of SPI-based film [19,25]. However, increasing the strength of SPI-based film without compromising its toughness remains a significant challenge.

The fiber reinforcement approach has been proven as a simple and effective way to improve the toughness of materials [1,2,26,27]. The improvement of toughness is achieved because fiber releases internal stress, and the interface between the fiber and the matrix consumes a certain amount of energy [28,29]. Common fibers can be categorized into synthetic fibers and natural fibers. Considering the complex process and the environmental pollution that occurs due to the use of synthetic fibers, natural fibers have received increasingly more attention from domestic scholars due to their advantages of biodegradation and natural regenerability [30,31,32]. Among these natural fibers, silk fiber (SF) is widely used in the textile industry due to its inherently high strength and excellent flexibility [33,34]. Previous research has confirmed that SF has a positive impact on the toughness enhancement modification of various composites, such as poly (butylene succinate), polyurethane, and polyethylene terephthalate [19,35,36,37]. However, the interfacial adhesion between SF and the matrix is mostly less reactive and weak intra/intermolecular hydrogen bonding.

Mussel-inspired dopamine surface modification has been widely used in the surface functionality of various substrates [17,18]. Dopamine (DA), with high concentrations of catechol and amine groups, can generate strong bonds on the surfaces of various inorganic and organic matrixes [9,11,38]. In addition, DA can generate self-polymerization under alkaline conditions (pH = 8.5), and the resulting polydopamine (PDA) layer is extensively used as a secondary reactive platform to react with the thiols and amine groups of the matrixes via Michael addition or Schiff base reactions [27,39].

In the present study, an eco-friendly, high-performance SPI/SF composite film was prepared. The introduction of dopamine coated-SF (PSF), together with WEU, was used to enhance the SPI film in order to improve its mechanical strength and water resistance ability, while not degrading its toughness. The surface compositions and microstructures of SF and PSF, and the thermostability, micromorphology, mechanical properties, and water resistance of the resultant composite films, were examined.

## 2. Materials and Methods

### 2.1. Materials

SPI (95% protein) was provided by Yuwang Ecological Food Industry Company, Qingdao, China. Silk fiber (SF) was obtained from ZheJiang University (Zhejiang, China) and processed into millimeters for further application. Dopamine (97% purity) and tris (hydroxymethyl aminomethane) were acquired from Tianjin Heowns Biochem Company (Tianjin, China). Bisphenol-A epoxy resin E44 was obtained from Shandong Yousuo Chemical Technology Co., Ltd. (Shandong China). Glycerol (99% purity) and other chemical reagents of analytical reagent grade were purchased from Beijing Chemical Reagents Company, Beijing, China. All chemicals were used without further purification.

### 2.2. Surface Modification of Silk Fiber by Dopamine Deposition

The SF was first dispersed into a mixture of ethanol and water (1:1) under vigorous stirring at 25 °C for 5 h to remove impurities, then dried at room temperature. The SF surface modification was facilitated simply by immersing the SF in a diluted dopamine buffer solution. The immersion solution (1.0 g/L) was pre-prepared by dissolving dopamine in a Tris-HCl (10 mM) buffer solution at pH 8.5 [18]. Next, 1.0 g of SF was added, and the mixture was then gently stirred at 30 °C for 6 h. The PSF was then purified through three cycles of pump filtration and distilled water rinsing, and then freeze-dried to acquire the dark brown fiber.

### 2.3. Preparation of Waterborne Epoxy Emulsions

E44 is an epoxy group-containing high molecular polymer, and it is difficult to cause depolymerization under normal conditions. In addition, the epoxy group on E44 can undergo an irreversible chemical reaction with the amino group on the molecular chain of the soy protein, which makes it more difficult to cause depolymerization of E44. The fabrication process of WEU is as following. First, E44 was dissolved in a mixture of 2-butoxyethanol and n-butanol via stirring in a four-necked round-bottom flask, and heated in an oil bath to 110 °C. The graft monomers (methacrylic acid, methyl methacrylate, and styrene) and initiator (benzoyl peroxide) were then dropped into the four-necked round-bottom flask using a constant pressure funnel for 2 h at 120 °C, and the mixture was then heated at 120 °C and stirred for 3 h. The introduction of methacrylic acid is to provide a hydrophilic group to the epoxy resin and to increase the reaction sites, and the introduction of styrene is to increase the hardness of the final film formation and the introduction of butyl methacrylate is intended to improve the flexibility and water resistance of the final film formation [21]. Finally, triethylamine was used to neutralize the mixture at 85 °C for 0.5 h, and then deionized water was added to form an oil-in-water emulsion with 10% solid content by vigorous stirring at 1200 r/min. The WEU was obtained after exhausting organic solvent (2-butoxyethanol and n-butanol) and partially unreacted styrene (the boiling point of styrene is 146 °C, and we think partially unreacted styrene could be removed by rotary evaporation) by rotary evaporation, and this WEU was used for the film preparation. The obtained WEU were extracted via the mixed solution of cyclohexane and absolute ethanol to obtain methacrylic grafted E44. After that, the grafted E44 was extracted by acetone twice. Pure grafted E44 was obtained and dried in a vacuum at 35–40 °C for 7–8 h for FTIR spectrum analysis and acid value determination [24]. The experimental details of WEU preparation are shown in Table 1.

### 2.4. Preparation of SPI-Based Film

The SPI-based film was prepared via a two-step casting method [16]. First, the SPI (5 g), glycerol (2.5 g), and distilled water (95 g) were sequentially added to a 250 mL beaker and stirred for 30 min. The beaker content was adjusted with an NaOH solution (10% (*w*/*w*)) to pH 9.0 ± 0.1, and then heated in a water bath at 85 °C for an additional 30 min. Certain amounts of SF or PSF and WEU (as listed in Table 2) were dispersed in the SPI solution while the mixture was constantly stirred. Then, the aforementioned suspension was poured into a Teflon tray with 150 mm in diameter after bubbles had been removed by an ultrasound treatment, and then vacuum dried at 60 °C to obtain SPI-based round film with a diameter of 150 mm. Films were stripped after drying for 24 h and placed in a saturated K_2_CO_3_-regulated (50% ± 2% relative humidity) desiccator for further use.

### 2.5. Characterization

Acid value was measured to investigate chemical grafting of E44. The theoretical acid value was calculated based on the amount of acrylic monomer [40]. The experimental acid value was measured as follows. E44 or WEU was weighed into a conical flask, and 50 mL of a mixed solvent (toluene/ethanol: 2/1), was added to completely dissolve the sample. The phenolphthalein indicator was added to the sample solution, immediately titrated with potassium hydroxide solution until red color appeared, and recorded the volume of potassium hydroxide solution consumed [41]. The experimental acid value (A, mg (KOH)/g) was determined as follows:$$A=56.1\times \frac{\left(V_{1}-V_{0}\right)\times C}{m\times NV}\times 100$$*V*_0_: The volume of potassium hydroxide solution consumed by the control group (mL); *V*_1_: The volume of the potassium hydroxide solution consumed by the samples (mL); *C*: Concentration of potassium hydroxide solution for titration (mol/L); *m*: Sample quantity (g); and *NV*: The non-volatile content of the sample (%).

Attenuated total reflectance-Fourier transform infrared spectroscopy (ATR-FTIR, Nicolet 6700, Beijing, China) in the wavelength range of 650–4000 cm^−1^ was employed to characterize the changes in the chemical structures of the SF, PSF, and SPI-based films with 32 scans.

The solid state ^13^C nuclear magnetic resonance (^13^C NMR, JEOLECS 400 MHz) spectra were used to investigate the chemical structural change of the SPI-based films, and were recorded with a 3.2 mm CP/MAS probe (Bruker, Beijing, China) at a spinning speed of 15000 Hz.

X-ray photoelectron spectroscopy (XPS, Thermo Fisher Scientific Co., Beijing, China) was conducted to analyze the surface chemical structures of SF and PSF with monochromatic Al Kα radiation (1486.6 eV). The X-ray beam was a 200 mm diameter beam raster over a 2 mm × 0.4 mm area on the specimens. Spectra were recorded using a pass energy of 50 eV and a resolution of 0.1 eV.

Scanning electron microscopy (SEM, Hitachi S-3400N, Beijing, China) was applied with an accelerating voltage of 5 kV to observe the surface morphologies of the SF, PSF, and SPI-based films.

Thermogravimetric analysis (TGA, Q50, Shimadzu, Beijing, China) with a temperature range of 40–600 °C and a heating rate of 10 °C·min^−1^ was performed under a nitrogen atmosphere (100 mL·min^−1^).

The mechanical properties of the SPI-based films were determined with a tensile testing machine (DCP-KZ300, Shimadzu, Beijing, China) at a loading speed of 50 mm·min^−1^ and an initial gauge length of 50 mm. The stress–strain curves of the specimens (10 mm × 80 mm) were obtained. The thicknesses of the films (five replicates) were measured with a digimatic micrometer. The tensile strength (TS) and elongation at break (EB) of each film were determined by a mean value of six replicates.

The water resistance of the SPI-based films was characterized by measuring their moisture content (MC), water absorption (WA), and total soluble matter (TSM) according to previous methods [7]. The surface hydrophilicity of the SPI-based films was investigated by water contact angles (WCA, OCA-20, Dataphysics Instruments GmbH, Beijing, China). A sessile droplet (about 3 μL, measured with a microsyringe) of distilled water was dropped onto the surface, and the angles of both sides were recorded at an interval of 0.1 s for 180 s.

## 3. Results and Discussions

### 3.1. Preparation of Waterborne Epoxy Emulsions

Epoxy crosslinking was an efficient approach for SPI-based film modification to improve its water resistance and mechanical properties. Active epoxy groups could induce ring-open reactions with the nucleophilic groups of the soy protein molecule to generate dense crosslinking networks, which is favorable for the performance improvement of SPI-based film. In this study, low-cost epoxy resin E44 with abundant active reactive sites was transformed into WEU to crosslink with the soy protein side chain. WEU were fabricated via radical grafting polymerization with hydrophilic monomers, and the grafting mechanism was shown in Scheme 1. E44 is an epoxy group-containing high molecular polymer, and it is difficult to cause depolymerization under normal conditions. In addition, the epoxy group on E44 can undergo an irreversible chemical reaction with the amino group on the molecular chain of the soy protein, which makes it more difficult to cause depolymerization of E44. The macroscopic dispersibility and stability of WEU and the successful grafting of hydrophilic groups onto the backbone of the epoxy resin molecule were investigated.

As shown in Figure 1b, WEU presented uniform dispersion after 15 days of storage, and no delamination was observed, indicating good stability. The dispersibility of the crosslinker is significant to the performance of SPI-based film. As seen from the digital images in Figure 1, the neat E44 presented obvious stratification in water (dyed by methylene blue), which indicates the poor dispersion of E44 in water, and seriously impedes the application of E44 in the hydrophilic SPI matrix. Compared with the neat E44, the WEU dispersed easily in water, and formed homogeneous and stable emulsions. The good dispersion of WEU in water can be attributed to the hydrophilic groups grafting on the molecule backbone of E44, as confirmed by FTIR test. As shown in Figure 1, the peak at 913 cm^−1^ was ascribed to the characteristic absorption of the epoxy group vibration peaks of the neat E44 epoxy resin. Peaks at 3250–3700 cm^−1^ were assigned to the characteristic absorption of hydroxyl –OH stretching vibrations of the epoxy resin molecule backbone. After the grafting of the hydrophilic groups, a new absorption peak appeared at 1723 cm^−1^ in the grafted E44 spectra. The experimental acid value (2.3 mg KOH/g) was close to the theoretical acid value (2.9 mg KOH/g) of WEU, indicating that the chemical reaction is a graft polymerization reaction of an unsaturated monomer with epoxy resin rather than a ring-open reaction between epoxy groups of E44 and carboxyl groups of methacrylic acid monomer. These results suggest that the hydrophilic groups (methacrylic acid) were grafted successfully onto the backbone of the E44 molecule. Moreover, the absorption band (3250–3700 cm^−1^) became broader because more –OH appeared in the WEU system, which indirectly indicates the successful grafting of methacrylic acid on E44. In conclusion, the hydrophilic groups were successfully grafted onto the E44 molecule backbone [21].

### 3.2. Surface Modification

The increase of crosslinking density always results in a decrease of toughness of SPI-based film. Therefore, natural SF was introduced into the SPI-based film to improve its toughness. However, the weak reactivity of the surface of SF has negative effects on the mechanical properties of SPI-based film. Therefore, mussel-inspired surface modification was used for the surface functionalization design of SF via a simple dip-coating approach. In this study, DA was used to coat SF and endow it with good reactive activity. The catechol in dopamine was oxidized to benzoquinone to form the PDA layer, which would then adhere onto the SF surface and could serve as a secondary reaction platform to induce multiple interactions. This mainly refers to the reaction between quinone and/or the –NH/–OH groups of the PDA layer and the –SH/–NH_2_ groups of the soy protein side chains via Michael addition or Schiff base reactions. FTIR, XPS, SEM, and TG measurements were performed to characterize the deposition of the PDA layer onto the SF surface.

The FTIR spectra are shown in Figure 2a, from which an obvious increase in the peak intensity of the region near 3275 cm^−1^ can be discerned. The peaks at 1511, 1444, and 1228 cm^−1^ are attributable to the –OH stretching vibration of aromatic hydroxyls, the benzene ring skeleton vibration, and the C–O stretching vibration of catechol, respectively. In addition, a new peak occurred at 1696 cm^−1^, which can be attributed to the C=O stretching vibration of benzoquinone [18]. 

The surface modification of SF was also ascertained by comparing the XPS spectra of the SF and PSF surfaces. As shown in Figure 2b, the SF exhibited three main peaks for C 1s, O 1s, and N 1s at 284, 532, and 400 eV, respectively. After coating by PDA, the peaks of the PSF at 284 and 532 eV significantly increased, while the peak at 400 eV decreased to 1.81% in accordance with the listed composition of chemical elements. From the XPS narrow scan of the C 1s region, it can be seen that the peak at 288 eV (C=O/N–C=O) almost disappeared, which might be attributable to the Schiff base reactions between the SF and PDA [9].

The thermal properties of the SF and PSF were examined by TGA, as shown in Figure 2c. The temperature at the maximal degradation rate in the DTG curve of PSF decreased to 292 °C as compared to that of SF, which was 308 °C. In addition, the residual weight of PSF slightly increased from 28.54% to 31.65% [32].

SEM measurement was also employed to prove the deposition of PDA. In Figure 2d, PSF shows a rough surface compared to SF after PDA depositing, and a thin and rough PDA layer was formed, as shown in Figure 2d [39].

These results indicate that the PDA layer successfully deposited onto the SF.

### 3.3. Structural Analysis of SPI-Based Film

In the present study, waterborne epoxy emulsions (WEU) and natural, highly reactive PDA-coated, SF (PSF) were applied to synergistically improve the mechanical performance of SPI-based film. Two main enhancement mechanisms were generated in the SPI-based film systems. First, WEU with multiple reaction sites were fabricated to crosslink the SPI-based film systems. A compact crosslinking network was generated via open-ring reactions between the epoxy groups in WEU and the side chains of soy protein. Second, PSF was used as an enhancement skeleton to release internal stress and improve toughness. The PDA layer also served as a secondary reactive platform to further crosslink with the side chains of the soy protein isolate via Michael addition or Schiff base reactions. The dual crosslinking network structure was confirmed by FTIR, XRD, and ^13^C NMR, and the detailed crosslinking mechanism is illustrated in Scheme 2, and the digital imagines are shown in Figure 3.

Figure 4a shows the ATR-FTIR spectra of the SPI-based films. The characteristic peaks at 1626, 1538, and 1260 cm^−1^ were respectively assigned to amide I (C–O stretching vibration), amide II (N–H bending), and amide III (C–H and N–H stretching) of the soy protein. The typical peaks at 3272, 2927, and 2875 cm^−1^ are attributable to O–H/N–H bending vibrations and the symmetric and asymmetric stretching of the –CH_2_ group, respectively [7]. As shown in Figure 3a, no obvious change presented between the neat SPI and the SPI/SF films, indicating a good biocompatibility between them. After the PDA surface modification of SF, the peak at 1260 cm^−1^ of the SPI/PSF film disappeared, partly indicating the chemical combination between the PSF and soy protein. For the SPI/WEU film, a new peak at 1509 cm^−1^ corresponded to the skeleton vibration of the benzene ring, and the typical peak at 1181 cm^−1^ belonged to the C–O–C group in the WEU. New peaks at 1236 and 827 cm^−1^ could have been caused by the unreactive epoxy groups. These results indicate that WEU were effectively inserted into the soy protein systems, and had excellent compatibility with the soy protein matrixes. After adding PSF, the intensity reductions of the peaks at 1236 and 827 cm^−1^ contributed to open-ring reactions between the epoxy groups of WEU and the soy peptide residues and suspended active groups of PSF, such as quinone and the –NH/–OH groups. According to the above results, we could conclude that an interpenetrating crosslinking network structure was formed by the internal physiochemical reaction between the WEU, PSF, and soy protein side chains.

XRD analysis was also employed to investigate the internal structure formed due to the interactions between the SF, WEU, and soy protein. As shown in Figure 4b, a peak at 20.3° was attributed to the β-sheet structures of the SF. Additionally, the characteristic peaks at 2θ = 9.2° and 20.2° respectively corresponded to the α-helix and β-sheet structures of the soy protein secondary conformation [17]. After introducing SF into soy protein matrixes, peaks at 29.0° of the SF disappeared, indicating interactions between the SF and soy protein [29]. In addition, the degree of crystallinity of the SPI/SF film exhibited an obvious increment compared to neat SPI film, which may have been caused by the high degree of crystallinity of the SF. Additionally, dopamine was developed to treat the SF surface to endow it with good reactivity. The degree of crystallinity of the SPI/PSF film decreased compared with that of the SPI/SF film, indicating that the crystalline structure of the SPI and SF was destroyed, which might be due to the internal crosslinking structure formed by the combination of the PSF and soy protein. This combination could appear to be a Schiff base reaction and Michael addition between the PDA layer and the soy protein side chain, such as –NH_2_ and –SH. In the XRD pattern of the SPI/PSF/WEU film, the peak at 9.0° of unmodified SPI/PSF film shifted to 8.5° after the addition of WEU, which suggests that the internal structure of the SPI-based film was further crosslinked by WEU via a ring-open reaction of epoxy with the soy protein side chain. In summation, the dual interpenetrating crosslinking network structure was formed in the SPI-based film due to various interactions between the soy protein, PSF, and WEU, which is consistent with the results of the FTIR analysis.

Furthermore, ^13^C NMR spectra were used to further detect the interactions between PSF, WEU, and soy proteins. Noticeable characteristic peaks in the neat SPI film spectra were the same as those in the results of previous research. Chemical shifts occurred at 15–25 ppm of methylene and methyl groups, 25–45 ppm of β-C, 45−65 ppm of α-C, 115−130 ppm of aromatic carbon, and 155–170 ppm of carbonyl. Compared to the neat SPI film spectra, a more intense chemical peak appeared at 177.3 ppm, which can be attributed to the amido linkages formed between the –COOH of the WEU and the soy protein side chain. In addition, compared with the neat SPI film spectra, a new peak formed at 157.5 ppm, which is consistent with the C=N linkages resulting from the Schiff base reaction between the benzoquinone of the PDA layer and the –NH_2_ of the soy protein. Compared with the neat SPI film spectra, the resonance strength related to the α-C in the SPI/PSF and SPI/PSF/WEU films became weaker, while the resonance strength related to the β-C in the SPI/PSF and SPI/PSF/WEU films became more intense, which denotes a covalent crosslinking reaction between the PSF, WEU, and soy protein matrixes. These results indicate that multiple intramolecular interactions were generated in the soy protein film systems, which would affect the mechanical properties of SPI-based films, as confirmed by FTIR and XRD analysis.

### 3.4. SPI-Based Film Micromorphology

The cross-sectional morphologies of SPI-based films were investigated via SEM to indirectly investigate the combination of the SF, PSF, WEU, and soy matrix. Figure 5 reveals that the pristine SPI film exhibited a relatively uneven fracture surface morphology and a special “river” pattern, consistent with results obtained by previous study [14]. Broad interspaces were found in the SEM image of the SPI/SF film, which confirmed the unfavorable interfacial adhesion between the SF and soy protein matrixes. However, the SPI/PSF film exhibited a favorable interfacial combination between the PSF and soy protein. This might be because the coated PDA layer generated multiple interactions, such as Schiff base reactions and Michael additions, with the soy protein side chain (–NH/–SH). Furthermore, we also found that the SPI/WEU film exhibited a relatively rougher fracture surface, which may be attributable to the crosslinking reaction between the epoxy groups of WEU and the soy protein side chain. The same result was found in SPI/PSF/WEU, in which the WEU served as a crosslinking agent and contributed to a more compact fracture surface. In particular, PSF was further tightly combined with the soy matrixes, indicating good bio-phase interfacial interactions between them. The surface imagine of SPI/PSF/WEU film indicated good dispersion of PSF in SPI matrix. These results indirectly indicate that the stable crosslinking skeleton formed by multiple interactions between the soy protein side chain, WEU, and PSF was tightly combined with the soy matrix.

### 3.5. The Thermal Stability of SPI-Based Films

TG analysis was conducted to measure the thermal stability of SPI-based film, and to investigate the effects of PSF and WEU on the thermal performance of SPI-based films. The TG analysis curves are provided in Figure 6, and the detailed data are presented in Table 3. As shown in Figure 5, the thermal degradation process of the neat SPI film can be divided into two stages. The first stage was mainly characterized by the evaporation of glycerol within the range of 120 to 270 °C, while the second stage from 275 to 550 °C was related to the thermal breakage of the soy protein backbone [18]. After pristine SF was introduced into the soy matrix system, the temperature corresponding to the maximum thermal degradation increased, indicating that the thermal stability of the SPI/SF film improved due to the good stability of SF itself. However, the thermal stability of SPI/PSF decreased compared to that of the SPI/SF and even the pristine SPI film. This may have occurred because, although the PDA layer provided more reactive sites for Schiff base reactions or Michael additions with the soy protein molecule, it also destroyed the surface structure of the SF, which is unfavorable to the thermal stability of SPI-based films. Moreover, the thermal stability of the SPI-based film further improved when the WEU crosslinker was added. As is evident from Figure 5, a new peak at about 370 °C appeared, which might be attributable to the thermal degradation of the benzene ring skeleton of WEU. We also noticed that the thermal degradation rates of the SPI/WEU and SPI/PSF/WEU films from 280 to 330 °C were lower than that of the neat SPI film, which indicates the improvement in thermal stability of the SPI-based films. In conclusion, the improvement in thermal stability of the SPI-based films benefited from two factors: The good stability of the WEU and SF themselves, and the stable crosslinking skeleton formed by multiple interactions between the soy protein side chain, WEU crosslinker, and PDA layer of the PSF, such as Schiff base reactions, the ring-opening reaction of epoxy, and hydrogen bonds.

### 3.6. Mechanical Properties of the SPI-Based Film

Figure 7a–c presents the mechanical properties of SPI-based films, including the thickness, Young’s modulus, and stress–strain curves and Figure 7d presents the possible physical and chemical combinations in SPI-based films. The thicknesses of the SPI-based films changed little compared with the pristine SPI film. After WEU were incorporated, the TS and Young’s modulus of the SPI/WEU increased to 5.56 and 81.28 MPa, respectively, which can be attributed to the crosslinking reaction between the active groups of the WEU crosslinking agent and the side chain of the soy molecule. The EB of the SPI/WEU film decreased to 55.35%, and contributed to the increase of the crosslinking density. After introducing SF into the soy film composites, the TS of the SPI/SF film decreased to 2.93 MPa. This may be due to the poor surface combination between the SF and the soy matrix. Dopamine was used to modify the SF and form a PDA layer on the SF surface. PSF exhibited better compactivity and more active reactivity with the soy protein matrix, resulting in increased values of TS of the SPI/PSF and SPI/PSF/WEU films, which were 6.25 and 6.91 MPa, respectively. The Young’s modulus of the SPI/PSF and SPI/PSF/WEU films also increased significantly, reaching 146.42 and 191.70 MPa and presenting 259% and 370% increments compared with pristine SPI film, respectively. In addition, the area value enclosed by SPI/PSF and SPI/PSF/WEU curves and the X axis was 4.59 and 4.49, which were larger than other curves, indicating better toughness of the SPI/PSF and SPI/PSF/WEU film. This may have been caused by the reduced stress concentration and crack propagation resulting from the PSF, which further toughened the SPI-based film. Therefore, the introduction of WEU and PSF has a positive impact on the mechanical properties of SPI-based film.

### 3.7. Water Resistance of the SPI-Based Film

The water resistance of different SPI-based films was determined by measuring their moisture content (MC), water absorption (WA), total soluble matter (TSM), and surface hydrophilicity (WCA). As shown in Figure 8, compared with pristine SPI film, the WA and TSM of SPI/WEU film decreased from 85.6% to 81.96%, and from 50.8% to 41.26%, respectively. This is because the WEU was physically/chemically combined with the soy side chain, as confirmed by the ATR-FTIR and ^13^C NMR results. After the introduction of SF, the TSM of the SPI/SF film increased from 50.8% to 63.47% as compared to the pristine SPI film, indicating a poor combination between the SF and soy protein matrix. After modification by dopamine, the PSF exhibited a better surface reactivity to induce multiple chemical bonds between the PDA layer and soy hydrophilic groups via Schiff base reactions and Michael additions. Compared to the pristine SPI film, the WA and TSM of the SPI/PSF/WEU film decreased to 75.21% and 37.28%, respectively, exhibiting significant decreases of 12.1% and 26.6%. Additionally, the WCA were measured to investigate the surface hydrophobicity of different SPI-based films, as shown in Figure 9. With the introduction of WEU, the WCA of the SPI/WEU film increased to 71.8° compared with the WCA of the pristine SPI film, which was 58.6°. In contrast, the WCA of the SPI/SF film decreased to 51.1° after introducing SF. The WCA of the SPI/PSF film increased to 75.8°, indicating good water resistance. After the synergistic modification of WEU and PSF, the WCA of the SPI/PSF/WEU film attained 76.5°. These results indicate that the water resistance of the SPI-based film improved significantly after the introduction of WEU and PSF, which might be due to the improvement of the SPI-based film crosslinking networks, as confirmed by FTIR, NMR, and XRD.

## 4. Conclusions

In this study, high-performance SPI-based film was fabricated by introducing water-dispersed, epoxy-based crosslinker WEU, and mussel-inspired, dopamine-decorated SF into the SPI matrix. As confirmed by FTIR, XRD, and ^13^C NMR, the epoxy groups of the WEU induced multiple reactions with the hydrophilic groups of the soy protein molecules to develop a dense crosslinking network, which endowed the modified SPI-based film with good water resistance and improved mechanical properties. Additionally, the PSF coated by the PDA layer presented favorable biocompatibility with the soy matrix, and also served as a skeleton structure to improve the mechanical performance of the SPI-based film.

The mechanical performance, especially the Young’s modulus, of the modified SPI-based film exhibited a significant improvement and increased up to 370% compared to the control. Simultaneously, the modified SPI-based film presented lower WC, MA, and TSM, while presenting higher WCA, indicating the improved water resistance of the modified SPI-based film. TGA testing indicated that the modified SPI-based film exhibited better thermal stability than the pristine SPI film. However, the toughness of the modified SPI-based film was still worse than the pristine SPI film after it was enhanced by PSF, which should be further addressed in future research. In summary, this study provides a simple and green method to develop high-performance SPI-based film composites with favorable water resistance and high Young’s modulus for application in food packaging, food preservation, and additive carrier fields.

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
