# Peer review of "High-Performance Soy Protein Isolate-Based Film Synergistically Enhanced by Waterborne Epoxy and Mussel-Inspired Poly(dopamine)-Decorated Silk Fiber"

_polymers, 2019, doi:10.3390/polym11101536_

Round 1

Reviewer 1 Report

The manuscript entitled ‘High-Performance Soy Protein Isolate-Based Film Synergistically Enhanced by Waterborne Epoxy and Mussel-Inspired Poly(dopamine)-decorated Silk Fiber’ by H. Pang et al. primarily discussed about the preparation and characterisation of Soy Protein Isolate (SPI) based composite film for food packaging application. The objective of this work is to enhance the water resistance and mechanical strength of the SPI based film to make it suitable for practical application. The authors blended SPI with modified epoxy resin and also loaded Poly(dopamine) functionalised Silk Fiber to form the composite film. Please see the comments below:

Firstly, preparation of waterborne epoxy emulsions, functionalisation of Silk Fiber by Poly(dopamine) and application of Silk Fiber for the reinforcement of Soy Protein Isolate-based composite are well known. So, combination of these prior art technologies is not really providing novel and exciting new science. Interestingly, some relevant references are missing in the manuscript, such as  

Bio-film Composites Composed of Soy Protein Isolate and Silk Fiber: Effect of Concentration of Silk Fiber on Mechanical and Thermal Properties, DOI: http://dx.doi.org/10.7234/composres.2014.27.5.196

Also, the references provided in the manuscript often misleading.

For example: Line 99: In this context, the cited reference #28 has no direct link with ‘Surface modification of silk fiber by dopamine deposition’.

Moreover, the research objectives were not properly validated. Results and discussions are not adequately revealed. For example,

Line: 43: ‘Common epoxy crosslinkers are mainly epoxy diluents, such as ethylene glycol, di-glycidyl ether and 1,2,3-propanetriol-diglycidyl-ether. Though the performance of SPI-based film is obviously improved with the introduction of epoxy diluents, their high cost and heterogeneous dispersion in matrix impede their practical large-scale application.’

Hence, one of the major objectives of this work is to develop a better materials system with competitive cost and better dispersion in SPI matrix. The current process for fabricating the composite film involves multiple steps such as modification of epoxy resin, functionalisation of Silk Fiber by Poly(dopamine) and then blending of these components in SPI. Also, 5g of modified epoxy system was used for 5g of SPI (Table 1) which is a significant proportion. As compared to epoxy diluents listed by the authors will this current system be still cost effective?

It’s not clear how thickness of sample 1, 0.264 mm (SPI: 5g) and thickness of sample 5, 0.281 mm (SPI: 5g + WEU 5g) are so similar (Table 1, Figure 6a). Images of the actual film samples and their superiority against related food packaging films were not compared.

There is no clear evidence about the quality of dispersion of modified epoxy system and Poly(dopamine) functionalised Silk Fiber in the SPI based composite film. The average dimensions of Silk Fiber, the strategy to overcome the possible agglomeration of the fibers in the composite film are not mentioned.

Line 280: 3.4. SPI-based film micromorphology:

One of the important aspects of the morphological study is to identify the matrix-fiber interfacial interaction in the composite film. The discussions around the morphological studies including interfacial interaction is mostly guess work. Quality of SEM images are not good enough and there is no direct evidence of the fiber dispersion within the matrix nor the details about matrix –fiber interphase.

Line 79: ‘In the present study, an eco-friendly, high-performance SPI/SF composite film was prepared.’

Some of the chemicals used in the current research are carcinogenic, such as (Line 102: Preparation of waterborne epoxy emulsions: Styrene, E44 (is this BPA – bisphenol A type epoxy resin?). There is always a risk of migration of incompletely polymerised or depolymerised BPA monomer. Such chemicals are not ideal for developing eco-friendly food packaging application.

Line 103: ‘First, E44 was dissolved in a mixture of 2-butoxyethanol and n-butanol via stirring in a four -necked round-bottom flask, and heated in an oil bath to 110 °C. The graft monomers (methacrylic acid, methyl methacrylate, and styrene) and initiator (benzoyl peroxide) were then dropped into the four-necked round-bottom flask using a constant pressure funnel..’

The modified epoxy resin is a major component in the composite film but there is no detailed information on the preparation and characterisation of this modified resin. For example, why three different monomers were used to modify the epoxy resin E44? What was the actual concentration/proportion of the epoxy resin, monomers, initiator and the solvent system?

The only information/characterisation about the chemical modification is the appearance of the peak at 1723 cm-1 in FTIR (Line 172). Full scientific details including changes in molecular weight of epoxy resin, reaction yield and evidence on how these monomers reacted with the epoxy resin are not explored.

Line 174: ‘Moreover, the absorption band (3700-3250 cm-1) became broader because more -OH appeared in the WEU system, which indirectly indicates the successful grafting of methacrylic acid on E44.’

The broad band in this region usually refer to hydrogen bonding and this could also happen due to the presence of moisture in the sample. There is no clarification on how more -OH could appear in the WEU system.

Line 109: The obtained WEU were extracted via the mixed solution of cyclohexane and absolute ethanol to obtain methacrylic grafted E44. After that, the grafted E44 was extracted by acetone twice. Pure grafted E44 was obtained and dried in a vacuum at 35-40 °C for 7~8 h for further investigation.

Line 117: Certain amounts of SF or PSF and WEU (as listed in Table 1) were dispersed in the SPI solution while the mixture was constantly stirred.

The primary discussions in the manuscript is based on waterborne epoxy emulsion (WEU) system, however, is it ‘grafted E44’ or WEU were actually dispersed in the SPI solution?

Line 165: Compared with the neat E44, the WEU dispersed easily in water, and formed homogeneous and stable emulsions.

There is no details/evaluation on the stability of this dispersion.  

Did the SPI and modified epoxy resin blend system form homogeneous phase in the dry film?

Line 172: The experimental acid value (93 mg KOH/g) was close to the theoretical acid value of WEU.

Details about the determination of experimental and theoretical acid values and how these values are useful in this context are not attempted.

Line 143: ‘The water resistance of the SPI-based films was characterized by measuring their moisture content (MC), water absorption (WA), and total soluble matter (TSM) according to previous methods.’

Methodologies for these experiments are not provided.

Line 209: ‘SEM measurement was also employed to prove the deposition of PDA. In Figure 2d, PSF shows a rough surface compared to SF after PDA depositing, and a thin and rough PDA layer was formed, as shown in Figure 2d.’

Authors did not try to investigate in details. For example, estimation of thickness and uniformity of the PDA layer on Silica Fiber. The actual fiber content in the composite film is very low (Table 1). So, how PDA coating on Silica Fiber could actually contribute in significant property enhancement, FYI below.

Line 182: The increase of crosslinking density always results in a decrease of toughness of SPI-based film[33]. Therefore, natural SF was introduced into the SPI-based film to improve its toughness.

The relative proportion of fiber content in the sample 5 composite film is only 0.02g in 5g SPI and 5g WEU. How crosslink density, toughness can significantly improve at this very low fiber content?

Figure 6a: How is it possible to significantly enhance the Young’s modulus in the SPI/PSF or SPI/PSF/WEU systems as compared to SPI/SF or SPI/WEU system?

Figure 8: How is it possible to substantial increase in the water contact angle of SPI/PSF system as compared to SPI/SF system while using a very low fiber loading? 

Hence, the research objectives and the practical purposes are not appropriately validated. The scientific details of this research and the interpretation of the results are not properly explored/discussed too. Overall, the manuscript is poorly written, not acceptable as per scientific standards of this field without having clear evidence of novelty and deep science study.

Reviewer 2 Report

The current manuscript entitled, "High-Performance Soy Protein Isolate-Based Film Synergistically Enhanced by Waterborne Epoxy and Mussel-Inspired Poly(dopamine)-decorated Silk Fiber" describes the new fiber polymers made of the waterborne epoxy emulsions (WEU), together with mussel-inspired dopamine-decorated silk fiber (PSF), and demonstrated that the obtained matrials synergistically improved the water resistance and mechanical properties. The work itself is interesting, and the manuscript is well-written. The data are promising, too. This reviewer agrees that this study could provide a simple and environmentally friendly approach to fabricate high-performance SPI-based film composites in food packaging fields. I would therefore, recommend the current manuscript to be accepted in the Polymers after a few minor revisions to be made as follows.

 Please include some other techniques used to prepare the similar or related fibers with some more references. What other possible application to be used, other than hose they already mentioned? Of there are, please specify. For the characterization, can the authors try the NMR technique or clearer verification? What about the XPS? If possible, this reviewer suggests to include.

Reviewer 3 Report

In my opinion, the new films obtained on base SPI by processing three components with different features, demonstrated a synergetic effect showed important features all of them in the final product. The soy protein isolation additionally processed with specifically prepared grafted epoxy resins together functionalized with dopamine silk fibers create the biodegradable film with interesting properties included water resistance and improved the toughness of the obtained material.

The experimental part contains a detail description of the processes of preparation of the materials. The physical parameters were characterized by various methods.

The minor points:

- the size of prepared films are not indicated

- the captions of Figures have to describe the content in more detail.

Reviewer 4 Report

Page 12. Line 349-352 “This may have been caused by the reduced stress concentration and crack propagation resulting from the PSF, which further toughened the SPI-based film. Therefore, the introduction of WEU and PSF has a positive impact on the mechanical properties of SPI-based film”

The decrease in elongation of SPI based films compared to neat SPI film is so high. Also, the increase in strength is not so high for SPI based films compared to neat SPI film. Then, how could you arrive at a conclusion for increased toughness or impact for SPI based films?

So the authors should either measure Izod impact strength test or Charpy impact test.

Or simply, they could show the stress strain curves in the manuscript and calculate the toughness from the area within curves and arrive at this conclusion.

Page 9 Line 285-286. How did the authors relate the relatively uneven fracture surface morphology and a special “river” pattern of SPI film to poor toughness? Toughness has the quantitatively measured.

You need to describe other factors like uneven distribution of various components leading to poor film formation…

It is always better to say neat SPI film – Avoids confusion for the readers when comparing to SPI-based films.

Page 3 Line 122 “Table Various SPI-based nanocomposite film formulations.”

How is this a nanocomposite? The cured or dried composites doesn’t have any components in nanoscale (<100 nm). The silk fibers are way high to micron scale. Use a different term for nanocomposite.

Round 2

Reviewer 1 Report

I appreciate the authors for their attempt to revise the manuscript and addressing the queries, however, I would recommend further improving this revised version to consider it suitable for publication. Please see the comments below:

Points 6, 15-18, response letter:

Section: 3.4. SPI-based film micromorphology

‘Poly(dopamine) coated silk fiber was prepared by a simple dip-coating method, in which dopamine would induce self-polymerization under alkaline environment. And the alkaline environment would cause degumming of silk fiber, which further form microfibrils. So prepared poly(dopamine) coated silk fiber has low density, and it also makes it more contact with soy protein and reacts with soy protein more easily. Finally, the performance of the SPI-based films was improved by the increase in crosslink density.’

The major claims for the enhancement in mechanical and water resistance properties of the films are largely depend on the interaction between Poly(dopamine) coated silk fiber and the soy protein matrix.

Ideally, not only the fiber-matrix interaction, the quality of dispersion of the fiber within the matrix should also play a crucial role. The SEM images provided in Figure 5 had identified single fiber within the matrix. I would suggest including images to get a broader picture of the quality of dispersion of silk fiber before and after polydopamine coating to address their impact on final properties.

Degumming of silk fiber could cause weight loss/lower the density of the fiber. However, there is no strong evidence to support that density of the silk fiber will still remain lower after polydopamine coating or the formation of microfibrils.

Point 7, response letter:

Section: 2.1. Materials

Line 90: ‘Commercial grade liquid epoxy resin (E44)’

-              Please include the actual chemical name of E44 (bisphenol A) in this section.

‘E44 is an epoxy group-containing high molecular polymer, and it is difficult to cause depolymerization under normal conditions. In addition, the epoxy group on E44 can undergo an irreversible chemical reaction with the amino group on the molecular chain of the soy protein, which makes it more difficult to cause depolymerization of E44.’

-              Please include the above explanation in section 2.3. Preparation of waterborne epoxy emulsions.

Point 11, response letter:

Section: 2.3. Preparation of waterborne epoxy emulsions

“Grafted E44” is a polymer after monomers was grafted onto the epoxy resin chain, while “WEU” is an aqueous dispersion of “Grafted E44”. And WEU served as waterborne epoxy crosslinker were dispersed in the SPI solution.

However, the query on whether ‘grafted E44’ or WEU were actually dispersed in the SPI solution for the preparation of SPI based film still need clarity.

Line 113: ‘The WEU was obtained after exhausting organic solvent and partially unreacted monomer by rotary evaporation. The obtained WEU were extracted via the mixed solution of cyclohexane and absolute ethanol to obtain methacrylic grafted E44. After that, the grafted E44 was extracted by acetone twice. Pure grafted E44 was obtained and dried in a vacuum at 35~40 °C for 7~8 h for further investigation.’

Line 124: ‘Certain amounts of SF or PSF and WEU (as listed in Table 1) were dispersed in the SPI solution while the mixture was constantly stirred.’

It seems that WEU (formulation provided in Table 1) was prepared first and grafted E44 was obtained from WEU. So, whether this purified grafted E44 was dispersed in water afterwards and that water dispersion was used for film preparation? In that case, purified grafted E44 dispersed in water cannot be ‘WEU’. Please clarify.

Line 113: ‘.. and partially unreacted monomer by rotary evaporation.. - please reword this statement to clearly address removal of partially unreacted monomer by rotary evaporation.

Point 13, response letter:

Line: 196: ‘The experimental acid value (93 mg KOH/g) was close to the theoretical acid value of WEU indicating that the polymerization reaction is a graft polymerization reaction of an unsaturated monomer with epoxy resin’.

Please provide the theoretical acid value of WEU and necessary details to support the determination of ‘theoretical acid value’. Also, elaborate how this experiment/acid value data support the ‘graft polymerization reaction’ providing further details.

Section 2.5 Characterization:

Please check the terminologies related to acid value equation and provide supporting reference for the equation to determine the acid value. Include the units in the equation.

Round 3

Reviewer 1 Report

Response to comment #1:

The digital images provided in Figure 5 are actually showing the presence of defects possibly due to the air entrapment occurred during film preparation. The fibers are not visible in these images and hence quality of fiber dispersion can not be revealed from these digital images.

Also, presence of these defects in the films would certainly impact on the end use/application, in particularly on the mechanical properties of the films. So, the claims for enhancement in mechanical properties due to chemical modification will remain uncertain if the control/modified film samples already contain voids/defects. I suggest to check this point carefully and replace these images with better ones, if possible.

Response to comment #4:

‘E44 is an epoxy group-containing high molecular polymer, and it is difficult to cause depolymerization under normal conditions. In addition, the epoxy group on E44 can undergo an irreversible chemical reaction with the amino group on the molecular chain of the soy protein, which makes it more difficult to cause depolymerization of E44.’

The above explanation given by the authors is still not included in section 2.3. Preparation of waterborne epoxy emulsions.

The above statement is also important to include in the text as ‘purified grafted E44’ was not used for the fabrication of SPI based films, rather WEU -  waterborne epoxy emulsion prepared using different monomers mentioned in Table 1, was dispersed into SPI solution.

One of the major aims of published papers is to provide adequate information to the readers so that if requires others can also repeat these experiments following the published protocol.

I suggest to include the followings:

Line 114: The WEU was obtained after exhausting organic solvent (2-butoxyethanol and n-butanol) and partially unreacted styrene (low boiling point)monomer by rotary evaporation.

Please include here: This WEU was used for the film preparation.

Line 117: Pure grafted E44 was obtained and dried in a vacuum at 35~40 °C for 7~8 h for further investigation.

Here, please provide the name of the investigations carried out using ‘Pure grafted E44’.

Response to comment #7:

‘The boiling point of styrene is 146 °C, and we think partially unreacted styrene could be removed by rotary evaporation.’

Please include this explanation in Line 115.

Line 145: m: sample quality (g);

It should be sample quantity.
